# Magnitude and factors associated with anemia among pregnant women attending antenatal care in Bench Maji, Keffa and Sheka zones of public hospitals, Southwest, Ethiopia, 2018: A cross -sectional study

Tesfaye Abera Gudeta[1]*, Tilahun Mekonnen Regassa[2], Alemayehu Sayih Belay[1]

1 Department of Nursing (Maternal Health Unit), College of Health Science, Mizan-Tepi University, Mizan-Aman, Ethiopia, 2 Department of Nursing (Adult Health Unit), College of Health Science, Mizan-Tepi University, Mizan-Aman, Ethiopia

* tesfeabera2013@gmail.com

## Abstract

### Background

Anemia during pregnancy is a common public health problem globally and it defined as the hemoglobin concentration of less than 11 g/dl. Anemia during pregnancy has maternal and perinatal diverse consequences and it increase the risk of maternal and perinatal mortality. The aim of this study is to assess magnitude and factors associated with anemia among pregnant women attending antenatal care in Bench Maji, Keffa and Sheka zones of public hospitals, South west, Ethiopia, 2018.

### Methods

A cross-sectional study was employed on 1871 pregnant mothers from selected hospitals. All third trimester pregnant women attending antenatal care at Mizan-Tepi University Teaching Hospital, Tepi, Gebretsadik Shawo and Wacha public hospitals were included in the study. Data was entered to Epidata version 3.1 and exported to SPSS version 21 for analysis. Logistic regression analysis was carried out to identify independently associate factors at confidence interval of 95% and significance level of P-value <0.05.

### Result

The magnitude of anemia in this study from the total study participant was 356 (19.0%). Among anemic pregnant women, 330 (92.7%), 21(5.9%) and 5(1.4%) were mild anemia, moderate anemia and severe anemia respectively. Age group 20–24 [AOR 6.28(2.40–16.42)], 25–29 [AOR = 6.38 (2.71–15.01)], 30–34 [AOR = 5.13 (2.27–11.58) and age ≥35 years [AOR = 2.53 (1.07–5.98)], educational status (read and write) [AOR 2.06, 95% CI (1.12–3.80)], gestational age(term)[AOR 1.94, 95% CI (1.27–2.96)], Caffeine (coffee and tea) and alcohol use occasionally [AOR 2.01, 95% CI (1.14–3.55)] and [AOR 2.59, 95% CI

**Data Availability Statement:** All relevant data are within the manuscript and its Supporting Information files.

**Funding:** The budget of this study was funded by Mizan-Tepi University.

**Competing interests:** The authors have declared that no competing interests exist.

**Abbreviations:** AOR, Adjusted Odds Ratio; ANC, Ante Natal Care; COR, Crude Odds Ratio; LNMP, Last Normal Menstrual Period; MUAC, Mid Upper Arm Circumference; SPSS, Statistical Package for Social Science; WHO, World Health Organization.

(1.49–4.52)] respectively, nutritional status (under nutrition) [AOR 3.00, 95% CI (2.22–3.97)] and family size (>6) [AOR 2.66, 95% CI (1.49–4.77)] were factors associated with anemia.

## Conclusion

The magnitude of anemia found to be high. Age, educational status of the mother, gestational age, caffeine and alcohol use, Nutritional status and family size were factors significantly associated with anemia. To prevent adverse outcome of anemia, health care providers should work on these factors.

## Introduction

World Health Organization (WHO) has defined anemia during pregnancy as the hemoglobin concentration of less than 11 g/dl [1]. Depending on hemoglobin concentration, anemia during pregnancy is classified as severe if the hemoglobin level is less than 7.0 g/dl, moderate when it falls between 7.0–9.9 g/dl, and mild from 10.0–11 g/dl [2–4]. The symptoms and signs of anemia are vague and nonspecific, including pallor, easy fatigability, headache, palpitations, tachycardia, and dyspnea. Angular stomatitis, glossitis, and koilonychia (spoon nails) may be present in long-standing severe anemia [5].

According to WHO, anemia is considered of a severe public health implication if its rate of ≥40% [6]. Anemia during pregnancy is a main public health problem worldwide, particularly in developing countries where there is inadequate diet and poor prenatal vitamins and iron and folic acid intake[7] and it affects the physical health and mental development of individual causing low productivity and poor economic development of a country[7,8].

Globally, every year anemia causes more than 115,000 maternal and 591,000 perinatal deaths [3]. Worldwide, anemia affects more than half billion reproductive age women [9–12]. It is the most common problem during pregnancy, therefore, 56% of pregnant women in low and middle income countries have anemia. Because of this reason, anemia during pregnancy contributes to 23% of indirect causes of maternal deaths in developing countries [8].

The prevalence of anemia was found be highest among pregnant women in developing countries, particularly in sub- Sahara Africa (57%), in South-East Asia (48%) and lowest prevalence (24.1%) was reported among pregnant women in South America [6].

Anemia in pregnancy has maternal and perinatal diverse consequences and it increase the risk of maternal and perinatal mortality [13, 14]. It also brings different obstetrical problems like; prematurity, low birth weight[15], abortion, intrauterine fetal death and perinatal mortality [16] and other maternal health problems like; impaired immune function, poor work capacity, fatigue, increased risk of cardiac diseases and mortality[8,14].

Even though there is different contributing factors for anemia like genetic, nutritional, and infectious disease factors, iron deficiency is the most common cause of 75% of anemia cases [8,17–20]. Iron deficiency anemia is common in pregnant women and it affects the development of the once country through decreasing the physical and cognitive development of children and productivity of adults [20].

The prevalence of anemia in pregnancy has remained unacceptably high and still it is a major public health concern in Ethiopia despite the fact that routine iron and folic acid supplementation during pregnancy was provided by the skilled providers [21]. This is due to the fact that poor nutritional intake, repeated infections, menstrual blood loss, and frequent

pregnancies are common in Ethiopia which is associated with poor socio economic conditions during pregnancy [22, 23] and poor antenatal care follows up during pregnancy [24].

In Ethiopia, about 17% of reproductive age women are anemic and 22% of them were pregnant [25]. Despite its known adverse effect on the pregnant women and children, there is no updated data available in the study area. Since no study was conducted in the study area, the finding of this study will be important to design appropriate interventions to reduce the high burden of the disease in the area and country at large. Therefore, this study is aimed at determining the magnitude of anemia in pregnant women and identifying its associated factors in the hospitals of Bench Maji, Keffa and Sheka zones Southwest Ethiopia.

## Methods and materials

### Study area and period

The study was conducted in public hospitals of Bench Maji, Sheka and Keffa zones namely, Mizan Tepi University teaching hospital (MTUTH), Tepi general hospital, Wacha hospital and Gebretsadik Shawo hospital from January 15- March 30/2018. MTUTH is located in Bench Maji zone on 560 kms from Addis Ababa. The two hospitals: Gebretsadik Shawo and Wacha hospitals are found in kefa zone at a distance of 441 and 520 kms away from Addis Ababa respectively, while Tepi general hospital is located in Sheka zone, 565 Kms away from capital city of Ethiopia, Addis Ababa.

### Study design

Facility based cross-sectional study design was used.

### Source and study population

All pregnant women who attending antenatal care at MTUTH, Tepi hospital, Gebretsadik Shawo hospital and Wacha hospital were considered as source of population and all pregnant women those fulfilled the inclusion criteria were considered as study population.

### Inclusion and exclusion criteria

All third trimester pregnant women attending antenatal care at MTUTH, Tepi, Gebretsadik Shawo and Wacha public hospitals were included in the study; however pregnant women who were critically ill and unable to communicate during data collection were excluded from the study.

### Sample size determination

The sample size was determined by using a single population proportion sample size calculation formula considering the following assumptions. d = margin of error of 2% with 95% confidence interval, estimated prevalence of anaemia is 23% [26] and considering non response rate of 10%. Then the final sample size became 1871.

### Sampling technique

All hospitals found in three zones were included in the study. The total sample size (1871) was allocated to the four public hospitals. The sample size allocation was based on the source of population from each hospital. The source of population of each hospital was taken from antenatal follow up report. Then the average was considered as source of population. The study participants were consecutively taken from each hospital until the sample size was achieved.

## Operational definitions and definition of terms

**Anemia in pregnancy**: In this study, anemia defined as hemoglobin level less 11 g/dl during third trimester. Woman with hemoglobin less 11g/dl was coded 1 whereas woman who was not anemic coded as 0.

**Pregnant women** are classified as non-anemic if hemoglobin $\geq 11.0$ g/dl, mild anemic if the range is 10 to 10.9 g/dl, moderate anemic if the range is 7 to 9.9 g/dl and severely anemic if hemoglobin is below 7.0 g/dl.

## Data collection instruments/tool

The data was collected using pre-tested questionnaire and anthropometric measurements. The questionnaire was developed based on tools that were applied in different related literatures (12–18). Questionnaires were developed in English and translated to Amharic by experts and translated back to English to see consistency of the question. The questionnaire contains sections for assessing anemia, demographics and associated factors.

## Data collectors

Twelve data collectors who bachelor degree holder midwives were recruited. Four supervisors who had master degree holders in maternal health were recruited.

## Data collection procedure

Data was collected through face to face interview, measurements and reviewing of medical record of the mother by using pre-tested structured questionnaire and check list by trained data collectors. Last normal menstrual period (LNMP) was confirmed from her chart and client report. Gestational age was calculated based on the last normal menstrual period (LNMP).

When LNMP-based gestational age is unknown, we relied on obstetric ultrasonography measures. Nutritional status was assessed by using Mid-upper arm circumference (MUAC) measurement. MUAC < 21cm considered as undernourished.

## Data processing and analysis

EPI data Statistical software version 3.1 and Statistical Package for Social Sciences (SPSS) software version 21.0 was used for data entry and analysis. After organizing and cleaning the data, frequencies & percentages was calculated to all variables that are related to the objectives of the study. Variables with P- value of less than 0.25 in binary logistic regression analysis was entered into the multivariable logistic regression analysis to control confounds so that the separate effects of the various factors associated with anemia could be assessed. Odds ratio with 95% confidence interval was used to examine associations between dependent & independent variables. P value less than 0.05 was considered significant. Finally the result was presented by using tables, charts and narrative form.

## Data quality control measures

The quality of the data was assured by using validated pre-tested questionnaires. Prior to the actual data collection, pre-test was done on 5% of the total study eligible subjects and have similar characteristics at Mizan health center and necessary amendments was made.

The validity of the tool was checked by face validity. Data collectors were trained intensively on the study instrument and data collection procedure that includes the relevance of the study, objective of the study, confidentiality, informed consent and interview technique. The data

collectors worked under close supervision of the supervisors to ensure adherence to correct data collection procedures.

Supervisors checked the filled questionnaires daily for completeness. Every morning, supervisors and data collectors conducted morning session to solve if there is any faced problem as early as possible and to take corrective measures accordingly. Moreover, the data was carefully entered and cleaned before the beginning of the analysis.

### Ethical considerations

Ethical approval was obtained from Mizan-Tepi University. Further permission was obtained from each hospital. After explaining the objectives of the study in detail, written informed consent was taken from all study participants.

## Result

### Socio-demographic characteristics

All the sampled mothers were participated (100% response rate). A total of 853(45.6%) participants were rural residents, 481(25.7%) were illiterates, 1808(96.7%) were married, 483(79.3%) were house wives and the family size of 1421(75.9%) participants were four children or less (Table 1).

### Variables related to obstetric characteristics

Around half 834 (44.6%) of the study participants were primigravida and almost all 1785 (95.4%) of the pregnancy were intended and also 1700 (90.9%) of the pregnancies were term pregnancy.

Majority 1726 (92.3%) of the participants have antenatal care (ANC) follow-up and only 424(25.1%) of participants were started antenatal follow up during their first trimester. And also the majority 1570 (83.9%) were take iron foliate during current pregnancy (Table 2).

### Variables related to pregnancy complication and medical illness

A total of 252(13.5%) of participants developed pregnancy-related complication during current pregnancy, 78 (31%), 31(12.3%),21 (8.3%), and 52 (20.6%) were developed preeclampsia, placenta previa, abruptio placenta and antepartum hemorrhage respectively. A total of 281 (15%) faced medical illness during current pregnancy. 1357(72.5%) women were not malnourished based on their mid-upper arm circumference (MUAC) measurement (Table 3).

### Variables related to behavioral factors

From the total study participants, 1516 (81%) used to drink caffeine (coffee and tea) on a daily basis, and 1272 (68.0%) were never drinking alcohol. Regarding the nutritional status 1252 (66.9%) and 1210 (64.7%) were get dietary counseling and additional diet during current pregnancy respectively (Table 4).

### Magnitude of anemia

The magnitude of anemia in this study from the total study participants (1871) was about 356 (19.0%) at **95 CI** (17.2%-20.7%). Among anemic pregnant women, 330 (92.7%), 21(5.9%) and 5(1.4%) were mild anemia, moderate anemia and severe anemia respectively (Fig 1).

**Table 1. Socio-demographic characteristics of women attending antenatal care in public hospitals of Benchi-Maji, Kaffa and Sheka zones, Southwest Ethiopia, 2018.**

| Variables | Category | Frequency | Percent (%) |
|---|---|---|---|
| Age | 15–19 | 168 | 9.0 |
| | 20–24 | 808 | 43.2 |
| | 25–29 | 547 | 29.2 |
| | 30–34 | 221 | 11.8 |
| | 35+ | 127 | 6.8 |
| Residence | Rural | 853 | 45.6 |
| | Urban | 1018 | 54.4 |
| Educational status | Unable to read and write | 481 | 25.7 |
| | Able to read write | 393 | 21.0 |
| | Primary education | 609 | 32.5 |
| | Secondary education | 246 | 13.1 |
| | College and above | 142 | 7.6 |
| Marital status | Married | 1808 | 96.7 |
| | Single | 36 | 1.9 |
| | Divorced | 5 | 0.3 |
| | Widowed | 10 | 0.5 |
| | Separate | 12 | 0.6 |
| Religion | Orthodox | 837 | 44.7 |
| | Muslim | 387 | 20.7 |
| | Protestant | 637 | 34.0 |
| | Other | 10 | 0.5 |
| Occupation | Housewife | 1483 | 79.3 |
| | Merchant | 170 | 9.1 |
| | Gov't employee | 117 | 6.3 |
| | Non-gov't employee | 18 | 1.0 |
| | Daily labor | 83 | 4.4 |
| Family size | ≤4 | 1421 | 75.9 |
| | 5–6 | 347 | 18.5 |
| | ≥7 | 103 | 5.5 |

## Factors associated with anemia

Mothers who in age group 20–24 [AOR 6.28 (2.40–16.42)], 25–29 [AOR = 6.38 (2.71–15.01)], 30–34 [AOR = 5.13 (2.27–11.58) and ≥35 years [AOR = 2.53 (1.07–5.98)] were more likely developed anemia as compared to younger age group (15–19). Mothers who have no formal education but read and write were two times more likely to have anemia as compared to mothers whose educational level of diploma and above [AOR 2.06, 95% CI (1.12–3.80)]. A pregnant mother who has gestational age ≥37weeks were two times more likely faced anemia as compared to preterm pregnancy [AOR 1.94, 95% CI (1.27–2.96)]. Pregnant mother who occasionally used caffeine (coffee and tea) and alcohol were two [AOR 2.01, 95% CI (1.14–3.55)] and two & half [AOR 2.59, 95% CI (1.49–4.52)] respectively times more likely developed anemia as compared to mothers never used this substance. Under nourished pregnant women were three times more likely developed anemia as compared to mothers who were well nourished in their nutritional status [AOR 3.00, 95% CI (2.22–3.97)]. Mothers who have larger family size (>6) were three times more likely faced anemia as compared to small family size [AOR 2.66, 95% CI (1.49–4.77)] (Table 5).

**Table 2. Variables related to obstetric characteristics among women attending ANC in public hospitals of Benchi-Maji, Kaffa and Sheka zones, Southwest, Ethiopia, 2018.**

| Variables | Category | Frequency | Percent (%) |
|---|---|---|---|
| Gravida | 1 | 834 | 44.6 |
| | 2–4 | 944 | 50.5 |
| | >4 | 93 | 5.0 |
| Parity | Primiparous | 883 | 47.2 |
| | Multiparous | 988 | 52.8 |
| Pregnancy status | Intended | 1785 | 95.4 |
| | Unintended | 86 | 4.6 |
| Gestational age | Less than 37 weeks | 171 | 9.1 |
| | ≥ 37 weeks | 1700 | 90.9 |
| ANC follow-up | Yes | 1726 | 92.3 |
| | No | 145 | 7.7 |
| Among mothers who have ANC follow up, At what month ANC started? | 1–3 months | 424 | 25.1 |
| | 4–6 months | 1185 | 70.0 |
| | 7–9 months | 83 | 4.9 |
| Number of ANC visit | One visit | 86 | 5.0 |
| | Two visit | 168 | 9.7 |
| | Three | 423 | 24.5 |
| | Four and above visit | 1049 | 60.8 |
| Iron foliate intake | Yes | 1570 | 83.9 |
| | No | 301 | 16.1 |

## Discussion

The world health organization estimates that the highest proportion of individuals affected by anemia are in Africa and also in Ethiopia anemia is a severe problem for both pregnant and non-pregnant women of childbearing age [6]. Therefore, this study was planned to assess the magnitude and associated factors of anemia among pregnant women.

The magnitude of anemia in the study area was 19.00%. The magnitude of anemia in pregnant women in this study area is higher than the study done in Addis Ababa (10.1%) [27]. The difference might be to socioeconomic difference, culture of dietary practice and awareness about anemia during pregnancy. The main cause of anemia in pregnancy is nutritional deficiency. So, giving attention during antenatal care about additional diet and supplementation of iron folate are very crucial in reducing the magnitude of anemia among pregnant mothers.

The magnitude of anemia in this study is lower as compared with the studies done in Malaysia (33%) [28], Gana (51%) [29], and in Ethiopia: Tigray(36.1%) [30], Nekemte town (52%) [31], Adama (28.1%) [32], Gode town (56.8%)[33], Bisidimo (27.9) [34], Jijiga town (63.8%)[35] and Ilu Abba bora zone (31.5%) [36]. This difference might be due to the study period and the attention given for focused antenatal care and supplementation of iron sulfate throughout the pregnancy.

The magnitude of this study is consistent with the studies done in India (20%) [37], Mekele town (19.7) [38], Mizan Aman general hospital (23.5%) [39] and Limo district (23%) [26].

In this study, factors influencing magnitude of anemia were identified. Advanced maternal age was statistically associated with anemia during pregnancy. The finding of this study is congruent with the studies done in Ghana and Jijiga [29, 35]. As maternal age increases, the mother may face pregnancy and labour related complications, and other illness which may predispose the mother for anemia. Mothers who haven't any formal education were more

**Table 3. Variables related to pregnancy complication and medical illness among women attending ANC in public hospitals of Benchi-Maji, Kaffa and Sheka zones, Southwest, Ethiopia, 2018.**

| Variables | Category | | Frequency | Percent (%) |
|---|---|---|---|---|
| Complications on current pregnancy | Yes | | 252 | 13.5 |
| | No | | 1619 | 86.5 |
| Pregnancy related complications | Gestational hypertension | Yes | 5 | 2 |
| | | No | 247 | 98.0 |
| | Preeclampsia | Yes | 78 | 31 |
| | | No | 174 | 69.0 |
| | Eclampsia | yes | 36 | 14.3 |
| | | No | 216 | 85.7 |
| | Placenta Previa | Yes | 31 | 12.3 |
| | | No | 221 | 87.7 |
| | Abruptio placenta | Yes | 21 | 8.3 |
| | | No | 231 | 91.7 |
| | Antepartum hemorrhage | Yes | 52 | 20.6 |
| | | No | 200 | 79.4 |
| Medical related illness on current pregnancy | Yes | | 281 | 15.0 |
| | No | | 1590 | 85.0 |
| Medical illnesses | Malaria | Yes | 156 | 8.3 |
| | | No | 1715 | 91.7 |
| | HIV | Positive | 51 | 2.7 |
| | | Negative | 1820 | 97.3 |
| | ART* status | Started | 51 | 100 |
| Nutritional status (Using MUAC*) | Under nutrition (MUAC<21cm) | | 514 | 27.5 |
| | Normal | | 1357 | 72.5 |

*ART = Anti-Retroviral Treatment

**MUAC** = Mid-Upper Arm Circumference

**Table 4. Variables related to behavioral factors among women attending ANC in public hospitals of Benchi-Maji, Kaffa and Sheka zones, Southwest, Ethiopia, 2018.**

| Variables | Category | Frequency | Percent (%) |
|---|---|---|---|
| Caffeine intake (coffee & tea) during index pregnancy | Never | 167 | 8.9 |
| | Daily | 1516 | 81.0 |
| | Weekly | 28 | 1.5 |
| | Occasionally | 160 | 8.6 |
| Alcohol intake during index pregnancy | Never | 1272 | 68.0 |
| | Daily | 27 | 1.4 |
| | Weekly | 86 | 4.6 |
| | Occasionally | 486 | 26.0 |
| Mother counseled on dietary practice during current pregnancy | Yes | 1252 | 66.9 |
| | No | 619 | 33.1 |
| Get additional diet during current pregnancy | Yes | 1210 | 64.7 |
| | No | 661 | 35.3 |
| Mothers faced physical harassment during current pregnancy | Yes | 129 | 6.9 |
| | No | 1742 | 93.1 |

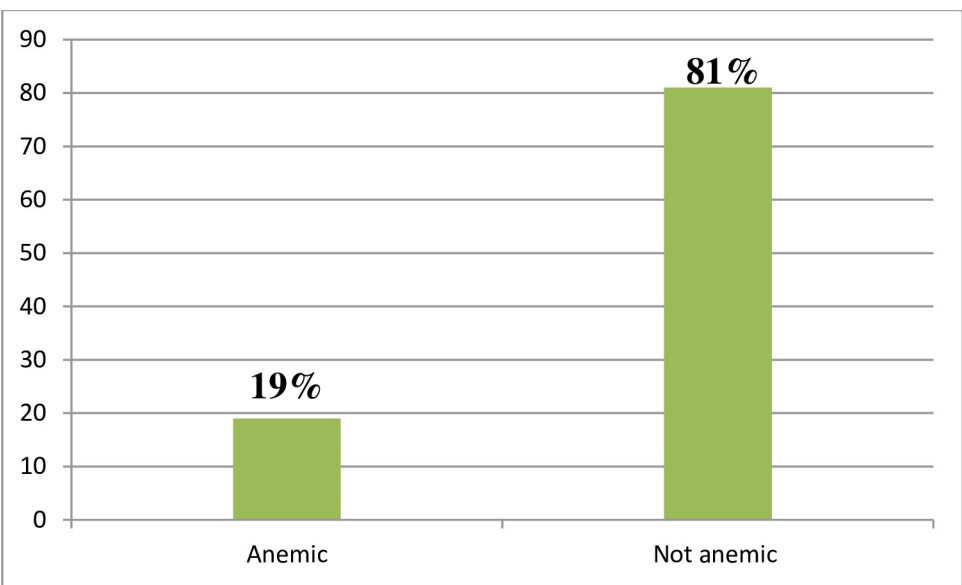

**Fig 1. Magnitude of anemia among women attending antenatal care in Bench Maji, Keffa and Sheka zone of public hospitals, Southwest Ethiopia, 2018.** This figure shows that the magnitude of anemia in pregnancy which was 19.0%.

likely develop anemia as compared to formally educated mothers. This finding is similar with study carried out in Malaysia and Tigray [28, 30]. It is obvious that as educational status increases, the life style, socio-economic status and diseases prevention knowledge and skilled also improved.

This study also identified other factors associated with anemia; gestational ages greater or equal to 37weeks were more likely faced anemia. Around 37 and above weeks, the demand of iron is increased which might be the cause for anemia. Mothers who have family size greater than six were more likely develop anemia as compared to mother who have less than five. This finding is consistent with the studies done in Jigjiga and Ilu Abba Bora zone [35, 36].

Pregnant women of Mid Upper Arm Circumference (MUAC) less than 21cm were more likely to be anemic as compared to women not malnourished. The result of this study is comparable with the study conducted in Adama town, Jigjiga town, Gode town and Ilu Abba Bora zone [32–36]. The similarity could be due the facts that under nutrition occur as a result of micro and macro nutrient deficiency and also anemia may occur as complication of malnutrition.

## Strength and limitation of this study

The study was carried out through close supervision and follow up during data collection period, and the analysis was generated from huge sample which increases its representativeness were considered as strength of the study. The study was facility based study; it is difficult to generalize for the community and the study might encounter inter observer error during measurements were considered as limitation of this study.

## Conclusion

The magnitude of anemia found to be high. Age, educational status of the mother, gestational age, caffeine and alcohol use, Nutritional status and family size were factors significantly

**Table 5. Factors associated with anemia among mothers attending antenatal care in public hospitals of Benchi-Maji, Kaffa and Sheka zones, Southwest, Ethiopia, 2018.**

| Variable | Category | Anemia | | COR (95% CI) | AOR (95% CI) |
|---|---|---|---|---|---|
| | | **No** | **Yes** | | |
| Age | 15–19 | 132 | 36 | 1 | 1 |
| | 20–24 | 649 | 159 | 0.90(0.60–1.35)0 0) 0 | **6.28 (2.40–16.42)** * |
| | 25–29 | 430 | 117 | 0.99(0.66–1.52) | **6.38 (2.71–15.01)**\* |
| | 30–34 | 188 | 33 | 0.64(0.38–1.09) | **5.13 (2.27–11.58)**\* |
| | 35+ | 116 | 11 | 0.35(0.17–0.71) | **2.53 (1.07–5.98)**\* |
| Residence | Rural | 647 | 206 | 0.54(0.43–0.69) | 1.37(0.98–1.92) |
| | Urban | 868 | 150 | 1 | 1 |
| Educational status | Cannot read and write | 355 | 126 | 2.30 (1.36–3.88) | 1.73(0.923–3.23) |
| | Read and write | 301 | 92 | 1.98 (1.16–3.38) | **2.06(1.12–3.80)**\* |
| | Primary education | 535 | 74 | 0.90(0.52–1.54) | 0.87(0.48–1.62) |
| | Secondary school | 201 | 45 | 1.45(0.81–2.59) | 1.70(0.89–3.26) |
| | Diploma and above | 123 | 19 | 1 | 1 |
| Parity | Primiparous | 740 | 143 | 1 | 1 |
| | Multiparous | 775 | 213 | 1.42 (1.13–1.80) | 1.40 (0.99–1.98) |
| Gestational age | Preterm (<37weeks) | 101 | 70 | 1 | 1 |
| | Term (> = 37 weeks) | 1414 | 286 | 0.29 (0.21–0.41) | **1.94(1.27–2.96)**\* |
| ANC follow up | Yes | 1432 | 294 | 1 | **1** |
| | No | 83 | 62 | 3.64 (2.56–5.17) | 1.56(0.90–2.71) |
| Intake Iron foliate | Yes | 1310 | 260 | 1 | 1 |
| | No | 205 | 96 | 2.36 (1.79–3.11) | 1.26(0.82–1.96) |
| Current pregnancy complications | Yes | 177 | 75 | 2.02(1.50–2.72) | 1.3610.94–1.98) |
| | No | 1338 | 281 | 1 | 1 |
| Mothers' HIV status | Negative | 1477 | 343 | 1 | **1** |
| | Positive | 38 | 13 | 1.47(0.78–2.80) | 1.82(0.89–3.72) |
| Caffeine intake (Coffee and tea) | Never | 136 | 31 | 1 | **1** |
| | Daily | 1256 | 260 | 0.91(0.60–1.37) | 0.69 (0.43–1.09) |
| | Weekly | 19 | 9 | 2.08(0.86–5.03) | 1.61(0.60–4.30) |
| | Occasionally | 104 | 56 | 2.36(1.42–3.93) | **2.01(1.14–3.55)** * |
| Alcohol intake | Never | 1073 | 199 | 1 | **1** |
| | Daily | 20 | 7 | 1.89(0.79–4.52) | 1.05(0.45–1.01) |
| | Weekly | 55 | 31 | 3.04(1.91–4.84) | 1.06 (0.38–3.00) |
| | Occasionally | 367 | 119 | 1.75(1.35–2.26) | **2.59(1.49–4.52)**\* |
| Counseled on dietary practice | Yes | 1020 | 232 | 1 | **1** |
| | No | 495 | 124 | 1.10(0.86–1.40) | 1.01(0.65–1.56) |
| Get additional diet during pregnancy | Yes | 993 | 217 | 1 | **1** |
| | No | 522 | 139 | 1.22(0.96–1.55) | 0.89(0.58–1.35) |
| Nutritional status | Well-nourished | 1171 | 186 | 1 | 1 |
| | Under nourished | 344 | 170 | 3.11(2.45–3.96) | **3.00(2.22–3.97)**\* |
| Family size | < = 4 | 1150 | 271 | 1 | 1 |
| | 5–6 | 291 | 56 | 0.82(0.60–1.12) | 1.054 .693 1.604 |
| | >6 | 74 | 29 | 1.66(1.06–2.61) | **2.66(1.49–4.77)**\* |

*(Continued)*

**Table 5.** (Continued)

| Variable | Category | Anemia | | COR (95% CI) | AOR (95% CI) |
|---|---|---|---|---|---|
| | | No | Yes | | |
| History of medical illness | Yes | 236 | 45 | 0.78(0.56–1.10) | 0.76(0.51–1.12) |
| | No | 1279 | 311 | 1 | 1 |

* = Statistically significant

AOR = Adjusted Adds Ratio, COR = Crude Odds Ratio.

associated with anemia. To prevent adverse outcome of anemia, health care providers should work on these factors.

## Supporting information

**S1 Table. Description of variables and measurement for the study in Bench Maji, Keffa and Sheka zones of public hospitals, Southwest, Ethiopia, 2018: This table shows that the description and measurements of dependent and some independent variables.** (DOCX)

**S1 File. Anemia SPSS data.** This SPSS data is a data which all statistical analysis was done from it. (SAV)

## Acknowledgments

We would like to express our deepest gratitude to our data collectors, supervisors, zonal health department, hospital directors and study participants for their valuable contribution in the realization of this study.

## Author Contributions

**Conceptualization:** Tesfaye Abera Gudeta.

**Data curation:** Tilahun Mekonnen Regassa.

**Formal analysis:** Tesfaye Abera Gudeta, Alemayehu Sayih Belay.

**Investigation:** Tesfaye Abera Gudeta.

**Methodology:** Tesfaye Abera Gudeta.

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
