## [Decision Letter · Decision Letter 0]

10 Sep 2019

PONE-D-19-15567

Magnitude and factors associated with anemia among pregnant women of Bench Maji, Keffa and Sheka zone Public hospitals, Southwest, Ethiopia, 2018: A cross sectional study

PLOS ONE

Dear Dr.Gudeta,

Thank you for submitting your manuscript to PLOS ONE. After careful consideration, we feel that it has merit but does not fully meet PLOS ONE’s publication criteria as it currently stands. Therefore, we invite you to submit a revised version of the manuscript that addresses the points raised during the review process.

We would appreciate receiving your revised manuscript by  9th October 2019. To enhance the reproducibility of your results, we recommend that if applicable you deposit your laboratory protocols in protocols.io, where a protocol can be assigned its own identifier (DOI) such that it can be cited independently in the future. For instructions see: http://journals.plos.org/plosone/s/submission-guidelines#loc-laboratory-protocols

We look forward to receiving your revised manuscript.

Kind regards,

Russell Kabir, PhD

Academic Editor

PLOS ONE

Journal Requirements:

2. Thank you for stating in your Funding Statement: "No -The funders had no role in study design, data collection and analysis, decision to publish, or preparation of the manuscript."

3. Please ensure that you refer to Figure 1 in your text as, if accepted, production will need this reference to link the reader to the figure.

Reviewers' comments:

Reviewer's Responses to Questions

**Comments to the Author**

1. Is the manuscript technically sound, and do the data support the conclusions?

Reviewer #1: Yes

Reviewer #2: Partly

Reviewer #3: Partly

Reviewer #4: Partly

Reviewer #5: Yes

Reviewer #6: Yes

2. Has the statistical analysis been performed appropriately and rigorously? 

Reviewer #1: Yes

Reviewer #2: No

Reviewer #3: Yes

Reviewer #4: No

Reviewer #5: Yes

Reviewer #6: No

3. Have the authors made all data underlying the findings in their manuscript fully available?

Reviewer #1: Yes

Reviewer #2: Yes

Reviewer #3: No

Reviewer #4: No

Reviewer #5: Yes

Reviewer #6: Yes

4. Is the manuscript presented in an intelligible fashion and written in standard English?

Reviewer #1: Yes

Reviewer #2: No

Reviewer #3: Yes

Reviewer #4: Yes

Reviewer #5: Yes

Reviewer #6: Yes

5. Review Comments to the Author

Reviewer #1: The author has been successful in addressing the research aim through research study design of facility based cross-sectional study and methodological soundness. The strength of the study is the sample size when considering several specific geographical locations around Ethiopia although selected hospitals have been taken into account. However, reviewing the article found me insufficiency of variables in terms of anemia and its magnitude. There could be more justified variables to address the issue more authentically. Furthermore, statistical analysis could be higher in terms of stating significant association between the variables with random effect so that there could be less bias in the study. References should be more contemporary to support the statements.

Reviewer #2: Edition with typology needs major revision

1. the introduction section need to have some sort of flow (what is anemia, the manifestation and how it is been diagnosed and then the magnitude of the problem and finally why you were interested studying on anemia n on the flow and the typology

2. In the method section, there was no description on how the study facilities were selected, and the study population was not stated correctly(those were sample population and not the study population because the were those from whom the study participants were selected

In addition the outcome variable should be clearly stated i.e. to which option was 1 given, was it for Yes or No, this is critical because your results are going to be written based on your outcome description

3. Result section: the analysis was not computed correctly and needs to be done again.

The discussion, and conclusion to be evaluated after reanalysis

Reviewer #3: Comment No 1 - Page 1:

Comment 1:

First:

In the Abstract, result section, arrange the percents in descending order.

Second:

Specify the older age group

Comment 2: Page 10

First:

Here, in this place you should state the magnitude of the research problem, and the study hypotheses, and the study objectives.

None of these are present.

Comment 3:Page 10

Add analytical, Facility based, cross sectional study.

Comment 4 , page 11:

Is it in proportionate to total population size in each of the 4 hospital.

Comment 5: Page 12 - line 8

midwives.

Comment 6: Page 12, line 24.

Rewrite multinomial Logistic regression

Comment 7: Page 14, line 1

You are studying 4 dimensions in relation to anemia among women as follows:

1-Socio-demographic characteristics. (Table 1).

2- Indicators related to obstetric characteristics. (Table 2).

3-Indicators related to pregnancy complication and medical illness.(Table 3) and,

4- Indicators related to behavioral factors. (Table 4).

So, you should cross tabulate each of these indicators with the anemia status among women, NOT to show them in frequency table.

It is better to construct 8 tables, after each of the cross tabulation table mentioned above, you have to construct the logistic regression table related to each of them.

Comment 8: Page 14, Line 3:

Rephrase the sentence to read like this:

The response rate was 100% where all the anticipated participants (1871) were participated in the study .

A total of 853(45.6%) participants were rural residents, 481(25.7%) were illiterates, 1808(96.7%) were married, 483(79.3%) were housewives and the family size of 1421(75.9%) participants was four children or less (Table 1).

Comment 9: Page 15, Table 2 :

It is better to show in a cross tabulation table presenting anemia status by obstetric characteristics among women..... instead of showing it in a frequency table.

Comment 10: Table 2:

Row (Gestational age) insert ≥ instead of >=37 weeks

Comment 11 - table 2:

Insert space (Among mothers who have ANC follow up, At what month ANC started? (4-6month) and (7-9month)

Comment 12, Table 2 :

Number of ANC visit - (left side alignment)

Comment 13,Table 2 :

Left side alignment (Iron foliate intake).

Comment 14 , Page 16 - Line 1

Change to: Indicators related to pregnancy complication and medical illness.

Comment 15 - page 16 - line 6:

Rephrase:

(13.5%) of participants developed pregnancy-related complication during current pregnancy, 78 (31%), 31(12.3%),21 (8.3%), and 52 (20.6%) were developed preeclampsia, placenta previa, abruptio placenta and antepartum hemorrhage respectively. A total of 281 (15%) faced medical illness during current pregnancy.

Comment 16 - Table 3:

It is better to show in a cross tabulation table presenting anemia status by pregnancy complication and medical illness among women..... instead of showing it in a frequency table.

Comment 17 - Table 3:

Rephrase:

Indicators of Pregnancy Complication and Medical Illness Presented among Pregnant Women Attending ANC in Public Hospitals of

Benchi-Maji, Kaffa And Sheka Zones, Southwest, Ethiopia, 2018.

Comment 18 - Table 3 - column 2:

Change category to indicators

Comment 19 , Table 3 - Center the table column heads.

Comment 20: table 3 - Last row - it is not necessarily to present started/not started) in a table, but if you would like to, make the corresponding value (not started) as (not applicable- not zero).

Comment 20, page 17:

Rephrase:

From the total study participants, 1516 (81%) used to drink caffeine (coffee and tea) on a daily basis, and 1272 (68.0%) were never drinking alcohol. Regarding the nutritional status1252 (66.9%) and 1210 (64.7%) were get dietary counseling and additional diet during current pregnancy respectively. 1357(72.5%) women were not malnourished based on their mid-upper arm circumference (MUAC) measurement.

Comment 21 - page 17,Table 4

Rephrase the title :

Distribution of Behavior Related Indicators among Pregnant Women Attending ANC in Public .......

It is better to show in a cross tabulation table presenting anemia status by behavioral factors among women..... instead of showing in a frequency table.

Comment 22 - Change column two head to indicators.

Comment 23, Last row, (Nutritional status using MUAC) this is not the suitable place to present this variable, because title of the table is related to behavioral factors among women attending ANC in public

hospitals. That , Nutrition status is not a behavioral factor. Table 3 could be a suitable place to show it since it presents the related to pregnancy complication and medical illness.

Comment 24 , table 4 - last row, change Under nutrition (MUAC<21cm) to Malnourished.

Comment 25, page 18 - line 1.

In table 4, You show a total of 514 respondents as being anemic , and here you state them to be 356 (19%); make the suitable corrections.

Comment 26, page18, line 2

These percents are not represent the (Among the anemic women) anemia; these percents are presenting the anemia categories ( severe , moderate and mild anemia respectively) among the total respondents. Make both corrections.

Comment 27:page 18, line 7.

Write it as OR (odds ratio), not AOR.

Comment 28, Table 5:

Delete column 3 (No, Yes) because none of the Odds Ratio analysis requires reference to this frequency.

Comment 29: Table 5, column 4,

First: It is not clear for me what you mean by the crude Odds Ration (COR).

Second:

Delete this column since you did not refer to this column content finding while discussion.

Third:

Better to delete this column and show the Beta (B) value instead.

Comment 30:Table 5

Rewrite Exp(B) - odds Ration instead of COR (In the last column)

Comment 40:

State your hypotheses first then continue with your discussion from there.

Comment 41: page 20 last line

You did not present any of the paper's [33 -37] main findings and/or conclusion in the introduction part. then, you show your discussion deferring to them. That will not help to see how they are related and/or differ from your study main findings and conclusion.

Comment 42 , page 21,

Where are the findings of these papers? It is the paper No. 24 that you lastly refer to in the introduction. Refer to the papers that you show in the introduction.

Comment 43 : page 21, (Strength and limitation of this study)

Strength of the study is how much it findings can contribute to stability and improvement of the pregnant women.

Reviewer #4: This cross-sectional study investigates the prevalence of and risk factors for anemia among 1871 pregnant women in South-West Ethiopian hospitals. The results show that 19% of the women had anemia and that anemia was associated with factors such as age, gestational age, family size, nutritional status, and caffeine and alcohol consumption. The authors conclude that health care providers should target these risk factors to reduce the problem of anemia during pregnancy.

The strengths of this study include the large sample size and the thoroughly-planned patient interviews. Although the manuscript is in general easy to understand, the grammar could be improved. My main concerns with the manuscript relate to the description of the recruitment process, the data presentation, and the interpretation the results. These concerns are detailed in the comments below.

Major comments

1. The manuscript does not contain information about the number of patients at each stage of the recruitment process. How many patients were screened for eligibility? (For example, how many women, regardless of their stage of pregnancy, received antenatal care at the four hospitals?) How many patients were excluded for ineligibility? I encourage the authors to consult the STROBE guideline.

2. The authors state that the response rate was 100%. This statement suggests that all eligible patients were willing to participate in the study. Is this true? Maybe the authors mean that they managed to recruit 1871 participants as planned?

3. The data collection procedure could be clarified. The authors explain that 12 midwives conducted the interviews and collected the data at 4 hospitals, but how were potential participants contacted and identified? For example, did the 12 midwives inform patients of the study during ordinary antenatal care visits? If so, were interviews conducted at the same visit or on a later occasion? Are the authors sure that the 12 midwives managed to identify all patients who potentially met the inclusion criteria, or could some patients have been missed because there were other midwives working at the hospitals?

4. The associations between anemia and its risk factors are examined using only odds ratios. As odds ratios measure relative effect instead of absolute effect, odds ratios do not necessarily convey the importance of risk factors from a public health perspective, which this study aims to do. The authors could fix this problem by including percentages in the third and fourth columns of Table 5, so that anemic and non-anemic patients can be easily compared.

5. The authors report the overall percentage of anemic patients at 4 hospitals. I encourage the authors also to report the percentage anemic patients by hospital, as this could differ.

6. The variable “Education status” is defined as a composite of both literacy (able/not able to read and write) and level of education (primary, secondary, or college). This definition is inappropriate because the categories are overlapping; a patient with a secondary education must be able to read and write. Even so, the results show that each patient is categorized according to only literacy or only level of education. The authors should explain this variable and probably redefine it. Furthermore, does level of education mean level of completed education?

7. Table 5 shows that a few variables (age, residence, and gestational age) have a reversed association with anemia after adjustment for confounding. Can the authors explain these results?

8. In Paragraph 1 of the discussion, the authors write that “…additional diet and supplementation of iron folate are very crucial in reducing the magnitude of anemia among pregnant mothers”. Although this may be common knowledge, the authors do not relate this statement to their own data. Perhaps they could comment on the fact that most anemic women in their study were receiving “additional diet”, were receiving iron and folate supplements, and were not undernourished.

9. The authors cite several previous studies of the prevalence of anemia. Were these studies conducted in hospital settings too?

10. In the discussion, the authors conclude that the observed 19% prevalence of anemia is high. However, the authors mention only one previous study that has shown a lower prevalence. As that study and the other studies that the authors cite were all conducted in Ethiopia or other low- to middle-income countries, the authors could further contextualize their results by relating them to the prevalence of anemia in high-income counties. The authors could also compare their results to the WHO’s definition of a severe public health issue, which the authors mention in the introduction.

11. The study has a few limitations that are not mentioned in the discussion. First, some variables are measured quite imprecisely, such as nutritional status, which is dichotomous and determined by upper-arm circumference. Second, some of the data are obtained in face-to-face interviews, which contain sensitive questions, so answers may not be reliable. Third, the blood tests do not specify the type of anemia.

12. The authors conclude that healthcare professionals can reduce the problem of anemia during pregnancy by targeting a number of risk factors: advanced age, advanced gestational age, education, caffeine and alcohol use, family size, and nutritional status. The authors should reconsider this conclusion for a number of reasons. First, the data do not actually show that advanced age is associated with a higher risk of anemia. Instead, the data show that teenage pregnancy is associated with a lower risk, although only after adjustment for confounding. This distinction is important, and it contradicts previous research (Adebisi & Strayhorn. Fam Med 2005;37(9):655-62), which should be discussed. Second, gestational age is not a modifiable risk factor, so it is unclear how healthcare professionals would work with this risk factor. Third, occasional consumption of caffeine or alcohol was indeed associated increased risks of anemia, but more frequent consumption was not clearly associated with anemia. This lack of dose-response relationships suggests that something other than caffeine and alcohol is driving the associations. Fourth, family size could be a socioeconomic indicator, reflecting income rather than biologic changes due to having had many pregnancies. In addition, there was only an association among women with a family size >6, who constituted only 5.5% of the study population.

13. The authors state that “[t]he funders had no role in study design, data collection and analysis, decision to publish, or preparation of the manuscript”. However, the authors do not mention who the funders are.

Minor comments

14. In the introduction, I encourage the authors provide contextual information about the study location. For example, how does South-Western Ethiopia relate to the rest of the country with respect to economy and demographics?

15. The authors recruited 1871 patients based on a sample-size calculation for estimating the proportion of women with anemia. Although this calculation is not wrong, it is superfluous because the authors do not use confidence intervals or p-values when analyzing the proportion of women with anemia. When confidence intervals and p-values are not used, there are no formulas to determine an appropriate sample size. Nevertheless, the decision not to use confidence intervals or p-values was a good one, because the authors recruited all eligible patients rather than a random sample.

16. The authors report that significant associations were observed between anemia and categorical variables with >2 levels, such as caffeine intake and alcohol intake, based on the p-value/confidence interval for individual odds ratios (Table 5). This method is common but considered incorrect in statistics because it increases the risk of a type 1 error. To avoid this error, the conventional method is to use likelihood ratio tests, which test whether at least one odds ratio is different from 1.

17. Should parity and family size be included in the same regression model (Table 5), due to the risk of collinearity?

18. In Table 1, “orthodox” could be clarified as orthodox Christian.

19. Does the variable “family size” refer to number of children or size of household?

20. The abbreviation ART is not defined in Table 3. Since ART status refers only to patients with HIV (instead of to all patients, like the other variables), this information could be placed in a foot note.

21. The results of the study would be easier to read if the five tables were combined into one. This could easily be done by replacing the “category” column in Table 5 with a “total” column for anemia status. The “category” column could then be incorporated into the “variable” column. In the second right-most column, crude odds ratios can be provided for all variables, even those that are not included in the multivariable analysis.

22. Figure 1 can be removed because it shows only the percentages of participants with and without anemia.

23. I interpret the variable “Get additional diet during current pregnancy” as the patient is eating more food than before the pregnancy. Is this interpretation correct? The authors may want to relabel this variable and ensure that the label is the same in both Tables 4 and 5.

24. The authors are correct that the generalizability of their results is limited by the fact that the study was hospital-based. However, it is not correct that generalizability is improved by a large sample size. A large sample size improves precision (reduces widths of confidence intervals and sizes of p-values).

25. Please paginate the manuscript.

Reviewer #5: As the author mentions- Gestational age was calculated based on the last normal menstrual period (LNMP)

It would be better to add 'using Naegele's rule' (https://ipfs.io/ipfs/.../wiki/Naegele's_rule.html) and give credit to

Franz Karl Naegele (1778–1851), the German obstetrician who devised the rule. Or it would be better to write the formula and cite it.

Few grammatical errors need to be corrected.

Reviewer #6: Data availability is clear

Ethical part is also clear

Should correct some spelling errors.

Should be clearon methodology,it is still not clear

Focus on sample size calculation

6. PLOS authors have the option to publish the peer review history of their article (what does this mean?). If published, this will include your full peer review and any attached files.

Reviewer #1: No

Reviewer #2: Yes: Dereje Birhanu (MPH), Assistant professor, Bahir Dar University

Reviewer #3: Yes: Ilham Abdalla Bashir Fadl

Reviewer #4: Yes: Jonathan Bergman

Reviewer #5: Yes: Lila Bahadur Basnet

Reviewer #6: Yes: Manita Pyakurel

---

## [Author Response · Author response to Decision Letter 0]

24 Oct 2019

We are revised all reviewers`comments and questions point to point. we uploaded our response to reviewers` comments and questions separately with other files.

---

## [Editor Report · Decision Letter 1]

30 Oct 2019

Magnitude and factors associated with anemia among pregnant women attending antenatal care in Bench Maji, Keffa and Sheka zonesPublic hospitals, Southwest, Ethiopia, 2018: A cross sectional study

PONE-D-19-15567R1

Dear Mr. Gudeta,

We are pleased to inform you that your manuscript has been judged scientifically suitable for publication and will be formally accepted for publication once it complies with all outstanding technical requirements.

With kind regards,

Russell Kabir, PhD

Academic Editor

PLOS ONE
---

## [Editor Report · Acceptance letter]

12 Nov 2019

PONE-D-19-15567R1 

Magnitude and factors associated with anemia among pregnant women attending antenatal care in Bench Maji, Keffa and Sheka zones Public hospitals, Southwest, Ethiopia, 2018:  A cross sectional study 

Dear Dr. Gudeta:

I am pleased to inform you that your manuscript has been deemed suitable for publication in PLOS ONE. Congratulations! Your manuscript is now with our production department. 

With kind regards,

on behalf of

Dr. Russell Kabir 

Academic Editor

PLOS ONE